# Research Progress of Gliomas in Machine Learning

**DOI:** 10.3390/cells10113169

**Published:** 2021-11-15

**Authors:** Yameng Wu, Yu Guo, Jun Ma, Yu Sa, Qifeng Li, Ning Zhang

**Affiliations:** Tianjin Key Lab of BME Measurement, Department of Biomedical Engineering, Tianjin University, Tianjin 300000, China; wuyameng@tju.edu.cn (Y.W.); guoyu@tju.edu.cn (Y.G.); majun1@tju.edu.cn (J.M.); sayu@tju.edu.cn (Y.S.); qfli@tju.edu.cn (Q.L.)

**Keywords:** gliomas, machine learning, prediction, radiomics, gene expression

## Abstract

In the field of gliomas research, the broad availability of genetic and image information originated by computer technologies and the booming of biomedical publications has led to the advent of the big-data era. Machine learning methods were applied as possible approaches to speed up the data mining processes. In this article, we reviewed the present situation and future orientations of machine learning application in gliomas within the context of workflows to integrate analysis for precision cancer care. Publicly available tools or algorithms for key machine learning technologies in the literature mining for glioma clinical research were reviewed and compared. Further, the existing solutions of machine learning methods and their limitations in glioma prediction and diagnostics, such as overfitting and class imbalanced, were critically analyzed.

## 1. Introduction

### 1.1. The Background of Glioma and Machine Learning

Glioma is a common primary central nervous system (CNS) tumor which originates from glial cells [1]. According to the classification scheme of the 2016 World Health Organization (WHO) Classification of Tumors of the Central Nervous System, adult diffuse gliomas consist of astrocytoma, oligodendroglioma, and glioblastoma. Gliomas can be divided into I to IV grades, in which stage I and II are low-grade gliomas, and stage III and IV are high-grade gliomas [2]. Grade IV gliomas are also known as glioblastoma multiforme (GBM), which is the most lethal brain cancer, and patients usually survive less than one year [3]. In the new 2016 WHO classification, each diagnostic category of diffuse glioma is defined by combining genotype and phenotype, which indicates the importance of molecular biomarkers, such as isocitrate dehydrogenase (IDH), mutational status in glioma has been recognized. With the improvement of computer computing speed and the development of computing methods, the Next Generation Sequencing (NGS) and imaging technology are developing vigorously. Genomics [4] and imaging information play an important role in helping scientists and physicians to understand the pathophysiological mechanisms in diagnoses and prognoses, and in choosing treatment plans. It is worth noting that the rapid development of technology has produced an enormous wealth of data, which can be used to infer the qualitative and quantitative relationship between DNA, RNA, proteins, and other cell molecules by mathematical and statistical tools. One of the main challenges of bioinformatics using NGS and image information is how to effectively transform medical big data into available knowledge. A key difficulty is that it takes a long time to process the large amount of data, which makes the traditional statistics-based algorithm less effective. To solve this problem, both industry and scientific research turn to machine learning methods for help.

Term machine learning refers to a series of processes that involves creating and evaluating algorithms that contribute to pattern recognition, classification, and prediction based on models generated from data. Because of its potential applications, machine learning methods are widely used in various fields, from pattern recognition [5], computer vision [6], spacecraft engineering [7], finance [8], entertainment [9], computational biology [10] to biomedical [11] and medical applications [12]. It is helping researchers process high-throughput data and image data more effectively and accurately.

Machine learning is a branch of artificial intelligence, which can imitate the computer algorithms of human intelligence. It is the computational algorithms that can automatically study experiences from data [13]. Machine learning algorithms build the model based on sample data, known as “training data”, to make predictions or decisions without being explicitly programmed to do so. It integrates and absorbs the ideas of different disciplines, such as artificial intelligence, probability and statistics, computer science, information theory, psychology, cybernetics, and philosophy [14,15]. The relationship between biology and machine learning has a long and complex history [16,17]. The early machine learning technology is called perceptron, which simulates the neurons of the human brain, and an artificial neural network (ANN) is produced [18]. Further development of artificial neural network structures, such as adaptive resonance theory (ART) [19] and neocognitron [20], are inspired by the human visual nervous system.

### 1.2. The Connections of Machine Learning and Glioma Research

Automated pattern recognition through machine learning is essential due to the enormity and complexity of biomedical data; manual analysis is both inefficient and untenable. Machine learning will help realize a future of knowing gliomas by unlocking the potential of large biomedical and patient datasets. It is known that machine learning could tackle medical tasks. Figure 1 shows the timeline of the combination of glioma and machine learning. Early uses of machine learning have shown promise to predict glioma biomarkers from MRI texture features [21], discover new biomarkers [22], classify tumor type [23], and fusion heterogeneous data from metabolic and molecular datasets [24]. Machine learning can assist biomedical scientists and medical professionals by identifying and summarizing meaningful patterns from gliomas’ large datasets [25].

Recognizing the increasing importance of understanding the pathogenesis of glioma, and the increasing reliance on machine learning for prediction, we believe that it is of great significance to review the published research on the prediction and prognosis of glioma of machine learning applications. The purpose of this review is to summarize over the past 20 years machine learning applications used in glioma. The purpose of this review is to statistic the literature on gliomas in machine learning over the past 20 years and summarize some typical research in recent years.

### 1.3. The Structure of This Article

This paper is divided into four sections. Section 1 presents the introduction of background in gliomas and machine learning methods and Section 2 gives an overview on the characteristics of machine learning and a literature review on the application of machine learning in gliomas. Section 3 analyzed several publications on machine learning approaches applied in gliomas over the past few years and further grouped them into three categories: the expression prediction of biomarkers, grade prediction, and prognosis of gliomas. The challenges with a few recommendations for future directions were presented in Section 4 and the last section is an overall discussion.

## 2. The Machine Learning Algorithms

### 2.1. Machine Learning Techniques

The machine learning algorithm is a branch of artificial intelligence research that builds and examines algorithms to boost pattern recognition, classification, and prediction. It uses various statistical, probability, and optimization tools to “learn” patterns from the large, noisy, or complex data sets, and then uses the prior training to classify new data, to identify novel patterns, or predict new trends [26]. Machine learning methods are similar to the methods that human beings usually use for learning; however it can draw a lot of energy from statistics and probability, fundamentally, it has more powerful functions because machine learning can carry out reasoning or decision-making, which cannot be achieved by using traditional statistical algorithms [27].

Supervised learning, unsupervised learning, and intensive learning are the three main aspects of machine learning. The typical examples of supervised learning are support vector machine [28], random forests [29], decision tree [30], etc., and the commonly used unsupervised learning algorithm is clustering [31]. It mainly uses supervised learning [32] and unsupervised learning [33] to deal with tasks in biology and medicine. Supervised learning predicts labels or classes on future data based on past data that includes labels/classes. Unsupervised learning identifies structure, usually clusters, amongst unlabeled data. 

In the research of gliomas and other tumors, when using machine learning technology to analyze omics data, the preprocessing methods are used to reduce the dimension of features. After this procedure, the input features needed by the machine learning algorithm are identified. Then the machine learning algorithm is used to build and test the prediction model. After that, the model based on input features can predict the output of tumor samples. Figure 2 shows this process briefly.

Collecting and integrating data from various channels is the first step in the machine learning process. In the scene of supervised learning, we also need to label the data. Data preprocessing mainly includes data normalization and data whitening. Machine learning can achieve the following goals: classification, regression, clustering, anomaly detection, and so on. The test set prepared in data preprocessing is used to test the model. After the evaluation of the model, parameter tuning will optimize the training model.

### 2.2. Machine Learning Methods and Gliomas

The occurrence and development of gliomas are usually associated with genetic abnormalities which can be revealed by gene expression data and imaging information and another supplementary way at the molecular level. The diagnosis and treatment of gliomas entail genomics and clinical medical imaging data. Novel computational methods such as machine learning have promoted the automatic extraction of important tumor markers for clinical treatment planning and post-treatment monitoring. For example, studies have shown that in some experiments of cancer survival prediction, the machine learning algorithm model performs better than the traditional regression model [34]. Machine learning methods were being extensively used in medical practice ranging from detection and prediction of tumors with CT [35] and MR images [23] to the identification of malignant tumors through proteomics [36] and genomics [37]. From the aforementioned cases, machine learning methods are particularly suitable for biological and medical tasks, especially those that rely on complex proteomic or genomic data. 

The machine learning techniques including decision tree and Naive Bayes have been widely used in the diagnosis and treatment of gliomas for decades [38,39]. In recent years, the amount of glioma research involving machine learning methods has increased dramatically. Owing to the open-access policies of many journals and the steady increase of scientific publications, the published papers are widely available. 

PubMed Central (Available online: http://www.ncbi.nlm.nih.gov/entrez/query.fcgi, accessed on 8 November 2021) is the U.S. National Library of Medicine’s web-based journal literature search system, which currently contains over 28 million citations from Medline, life science journals, and online books (PubMed). The number of publications each year, as retrieved in PubMed, has surpassed one million since 2011. Figure 3 shows the quantity and trend that we found in PubMed Central by using the keywords shown in Appendix A by year (from 2002 up to 2021). The correlation of the unique papers was evaluated by reading the titles and abstracts and recognizing papers that used identifiable machine learning methods, as well as molecular, clinical, histological, physiological, or epidemiological data in carrying out a glioma prognosis or prediction. The chosen papers covered decision trees, Naive Bayes, random forests, minimum Redundancy Maximum Relevance Feature Selection (mRMR), k-nearest neighbors (k-NN), Convolutional Neural Network (CNN), Support Vector Machine (SVM), and other algorithms. No restrictions were implemented in the resulted hits except the exclusion of papers published before 2000. Based on the recent PubMed results concerning the subject of machine learning and gliomas more than 171 articles have been published thus far. The overwhelming majority of these publications employed one or more machine learning algorithms and merging data from the heterogeneous data source for the identification of gliomas as well as for the treatment and prognosis of gliomas. 

As mentioned above, the number of publications presented in Figure 3 refers to the accurate numbers retrieved from the databases without any modification. It can be observed from the picture that the enthusiasm of researchers in this particular field has increased exponentially since entering the 21st century, and in the past three years, research heat has increased the fastest.

## 3. A Survey of Machine Learning Applications in Gliomas

MR images and histopathology are the most important indicators in the determination of whether it is gliomas or other abnormalities. Machine learning methods have good performance at pattern recognition of images. Gene mutations are essential in the determination of tumor molecular grade. Uncovering more genes related with gliomas may help us to know the pathway of gliomas. Machine learning methods are good at processing high-dimension gene expression profiles. Therefore, the problems involving gliomas radiograph, gene expression, and histopathology often tend to be tackled by machine learning methods.

Most of the glioma biomarkers were tested by gene detection, gene methylation detection, and an immunohistochemistry (IHC) test. The diagnosis of glioma grades often requires not only pathology but also molecular biomarkers. Traditional machine learning methods have greater advantages in computing speed, computing scale, and cost-savings in mining omics data. Hence, less complicated methods such as decision tree, Naïve Bayes, SVM and KNN are better choices. 

Treatment decisions are based on histopathological diagnosis and grading. To understand more accurately about the tumor, IDH mutation, 1p19q deletion, and MGMT status tests are recommended in general. In some cases, TERT, EGFR, BRAF V600E and H3K27 tests are also needed. Some other biomarkers are regarded as the common glioma protein targets, such as Ki67, S100 and GFAP, which may not be solid. Their roles in gliomas are still under investigation and controversies have been observed in experiments [40].

Based on our analysis of the most recent studies on gliomas machine learning models, we categorized the current publications according to their functionality into three main aspects: (i) predicting the expression of biomarkers; (ii) cancer grade prediction; and (iii) the survivability risk prediction. The first case means the application of machine learning methods in predicting the status of IDH, 1p19q, MGMT, and other biomarkers in gliomas. The second case reviewed the computing methods for predicting glioma grades, and the third case tries to predict outcomes after the diagnosis of gliomas, such as life expectancy, survivability, progression, and tumor drug sensitivity.

In this perspective, we outline a vision for how machine learning can be applied to make critical advances in gliomas. We focus on those three areas in the next three subsections. Though there are some areas which have a relatively small number of studies, we do not discuss them in this review (Table 1). We reviewed several chosen papers published in recent years in those three aspects, which are typical in solving glioma clinical issues, data processing, and model building, by presenting the adopted techniques and the proposed model configuration. Table 2 depicts some of the publications presented in this review.

### 3.1. The Expression Prediction of Biomarkers in Gliomas

The findings of pathological results are the premise of rational treatment. Presently, the malignant and benign tumors and molecular subtypes are determined by pathological examination during surgery or biopsy [56]. Then, the following immunohistochemistry (IHC) test determines the molecular biomarkers of tumor tissues at the microscopic level. These pathologic biomarkers, typical proteins, genes, and other biomarkers, are useful indicators for diagnosis, prognosis, or treatment response [57]. However, obtaining such information for gliomas requires invasive approaches. Surgical decision-making could be difficult and time-consuming for many patients. Those patients who are not eligible for surgery or seek non-surgical treatment may have limited treatment options without pathological guidance. Therefore, presurgical glioma status and the expression of biomarkers are valued and preferred with non-invasive approaches. In recent years, many studies have reported that it is possible to predict the expression of biomarkers in gliomas from pre-operative medical images by using machine learning [58].

This section aimed to discuss the research using machine learning methods on the prediction of the expression of biomarkers in gliomas. Based on our survey, we here present the most relevant and recent publications that proposed the use of machine learning techniques for biomarker expression predicting. A work using random forests regression algorithms to predict the glioma pathologic biomarkers and tumor grades on gene expression profile is commented on by [48]. It is a typical study among the other published works in the glioma detection field using gene expression profile. In their work, random forests frequently used machine learning models were employed to identify potential biomarkers related to glioma survival. Only utilizing histological information in studying various types of gliomas is restricted. Previous studies have shown that gene expression profiling provides an objective method to classify tumors. Machine learning algorithms are good at mining patterns from large-scale gene expression profiles. The authors claimed that their machine learning-based approach can identify 104 genes which can be used as core genes related to patient survival. Ten genes can potentially serve as indicators to classify patients with gliomas into different risk groups and could be used to estimate the prognosis of patients with gliomas. 

In clinic, the detection of glioma biomarkers is always through gene detection or immunohistochemistry test, while some research tends to predict the status of these biomarkers by medical images. MRI has proven to be indispensable for brain tumor imaging and most research focused on conventional MR sequences for evaluation. Except Ki67, GFAP and S100, other glioma biomarkers, such as ATRX, IDH1 and 1p/19q statues can also be identified by medical imaging. In Haubold et al. research [49], the authors enhanced the imaging platform for quantitative and radiomic analysis by introducing MR fingerprinting as an additional MR sequence. In the study, linear SVM and random forests were used for prediction of the mutational status of ATRX, IDH1, and 1p19q of patients with cerebral gliomas, which were based on data from multiparametric ^18^F-FET PET-MRI. A training set was used for 3-folds cross-validation training with 20 repeats, each of which was trained on a different subset of the samples to provide for a robust and diverse committee of linear SVM and random forest, while pooling across the repeats of the predictions on the testing folds. Compared with the research which use conventional MRI [59], this study used more advanced MR fingerprinting as an additional MR sequence and it resulted in a better performance. The publication showed that MRI can perform well in the prediction of expression of biomarkers in gliomas.

Another interesting article also published in 2020 [50] proposed a neural network model that used multimodal data including MRI, positron emission tomography (PET), and computed tomography (CT) for the prediction of mutations in the IDH gene and the codeletion of chromosome arms 1p and 19q (1p19q). We should highlight an important aspect of this work regarding using multi-modality data to directly predict the groups of molecular expressions. Residual networks were employed in this study to extract features followed by training a committee of 217 models using leave-one-out cross-validation. It was found that the deep neural network could accurately predict IDH mutation and 1p19q codeletion when the MRI, PET, and CT data were combined. The authors claimed that the model was trained only according to low-grade gliomas data, which could be useful when other researchers applied the method on glioblastomas. This procedure of establishing different distinct molecular subtypes could be useful in distinguishing low-grade gliomas (LGG) and glioblastomas. However, the accuracy is a little lower than just using MRI as the results of using imaging fusion technology do not show advantages in the accuracy of biomarker identifying.

### 3.2. Glioma Grades Classification

Treatment options and responses differ from diagnosis [60]. The significance of the grading system is to mark the likely growth rate of the tumor and the possibility of tumor spreading in the brain, which can be used to predict curative effects and carry out the treatment plan. The grade and classification of brain tumors are formulated by the WHO. Glioma staging covers different types of tumors, many of which have significant differences in biological characteristics, prognosis, and treatment.

According to the latest WHO guidelines for the diagnosis of gliomas, the diagnosis of gliomas should include not only pathological indicators, but also the status of molecular biomarkers. Some researchers are now exploring whether machine learning can simplify the process of glioma grading. A work that studied multimodal MR radiomics that can stratify gliomas’ molecular subtypes is proposed in [51]. They suggested a three-level machine learning model composed of four binary classifiers to stratify five molecular subtypes of gliomas. They exploited multimodal MR radiomics to provides a reliable alternative to determine the pathology and molecular subtypes of gliomas. The approach proposed by the authors is comprised of two parts, the classification model-building part, and the performance evaluation part. The classification model building phase is itself comprised of six SVM and three ensemble learning approaches. The algorithm SVM is ideal for glioma prediction, which is easy to control the complexity of decision rules and the frequency of errors, as well as overfitting is unlikely to occur. The decision tree is another popular choice, as it results in a more human-friendly structure that can provide an understanding of how the system makes a choice. This work employed advanced multimodal MR radiomics to construct more comprehensive functional and metabolic radiomics in the characterization of gliomas. The authors claim that their model can effectively stratify five molecular subtypes to benefit the diagnosis and monitoring of gliomas. If gliomas molecular grades can be classified only through medical images, there will be less burden of patients and more efficiency of the hospital process.

Some of the researchers are concerned about the effect of supervised feature selection. Feature selection is necessary as the data usually contain many irrelevant, redundant, and noisy expressions. Effective data engineering can avoid the “garbage in, garbage out” consequence in machine learning problems. For example, Sengupta et al. [52] proposed an optimized SVM classifier to handle the problems of glioma grading using T1 perfusion parameters and volume of tumor components. The authors applied random forests to obtain optimal features for building an SVM classifier, which provided better grading results than the above result. The classifier was evaluated using 12-fold cross-validation and can achieve satisfactory classifications with an error of 3.7% for grade II vs. III, 5.26% for grade III vs. IV, and 9.43% for Grade II vs. III vs. IV. After effective data engineering, the performance was better than the last research. A potential limitation in this study was the absence of Grade I patients. The model could likely be further optimized by eliminating data imbalance between groups. In this research, most of the methods require manual tumor delineation, which is one of the limitations of this kind of experiment.

In the scenario of glioma prediction, it is common to occur data imbalance. Some research solved this issue and reached a more reliable result. The advantages of the Naive Bayes algorithm is not only easy to understand and can efficiently train, but also can be founded based on statistical modeling, solving the problem of data imbalance between groups was also effective. Niu et al. [53] employed five machine learning methods in a study aiming to predict the glioma stages based on the selected key genes. A total number of 527 gene expression data of brain tissue samples of Homo sapiens downloaded from the GEO (Available online: http://www.ncbi.nlm.nih.gov/geo/, accessed on 8 November 2021) database were considered in this study. A specific classification model was followed with the employment of two algorithms, namely random forests and Complement Naive Bayes. As a result, the prediction accuracy by using random forests was 97.1% for Grade I-II, and the prediction accuracy by using Complement Naive Bayes was 72.8% for Grade III-VI; any bias could be avoided when building the most effective model of imbalanced data. In their study, the authors selected 19 genes between Grade II glioma through bioinformatics method and protein-protein interaction network followed by the random forests. They filtered 21 genes between Grade III glioma and Grade II through bioinformatics method and protein-protein interaction network followed by the Complement Naive Bayes algorithm. The authors combined the more accurate gene expression data in order to yield a comprehensive insight into the molecules involved in the pathogenesis of gliomas.

### 3.3. The Prognosis Prediction of Gliomas

The incidence rate, cancer recurrence, and cancer survivability are three important cancer prognosis predictive facets. The first scenario predicts the possibility of developing cancer before the disease occurs. The second scenario tries to predict the likelihood of redeveloping cancer and the third scenario predicts the outcomes after the disease diagnosis, such as life expectancy, survivability, progress, and tumor drug sensitivity [61].

The term survival refers to the time interval from the beginning of a patient to the occurrence of an event, such as the period of the beginning and end of a recovery, or the time from the cancer diagnosis until death [62]. In medical research, survival analysis is often used to evaluate data from time-to-event. Survival analysis is a different field concerned with predicting the time until a medical condition occurs. From the perspective of machine learning, survival analysis is a ranking problem in which data points are ranked on their survival times rather than predicting the actual survival times [63]. Oncologists often face the difficult tasks of predicting the prognosis and survival of patients with refractory malignant tumors [64]. Their assessments in these cases are based on clinical experience and comprehensive knowledge of patients. However, physicians usually tend to overestimate the survival risk of patients with advanced cancer, as such predictions are largely unreliable, inaccurate, and generally more optimistic. Survivability prediction is related to several predictive facets that consist of genetic factors, size, as well as grade and stage of the tumor. When physicians have a good understanding of the prognosis of patients, patients are likely to receive more accurate treatment [65]. Necessity is apparent for physicians to have the ability for formulating a correct estimation of survival among patients with advanced and incurable cancers in the medical decision-making process, which involved data analysis, classification, and prediction.

Some research merged multi omics data of gliomas and tried to mine more information for glioma prognosis. An interesting article published in 2018 [54] proposed a convolutional neural network-based model with traditional survival models from pathology images. It can tolerate noisy inputs as well as be used for both regression and classification, which is suitable for the prediction of prognosis. The authors advocate the idea that the treatment planning for gliomas is dependent on many factors, including patient age and grade, which have been limited by considerable intra- and inter-observer variability. The article integrated the microscopic images of tissue biopsies and genomic biomarkers into a single prediction framework. They provided a comprehensive insight into the molecules involved in the pathogenesis of glioma. The authors claimed that their approach surpasses the prognostic accuracy of human experts using the current clinical standard for classifying brain tumors. They presented an innovative approach for objective, accurate, and integrated predictions of patient outcomes. Further analysis is still required to combine pathology images with rich genomic and clinical annotations to clarify the mechanisms underlying glioma tumor genesis and development.

A work by [55] developed a signature associated with the tumor immune microenvironment using machine learning. The study showed the improvement in predicting the survival rates of glioma patients by using LASSO Cox regression algorithm. Based on biological knowledge, the authors investigated the immunogenomic landscape of glioma followed by developing an immune-relevant prognostic 15-gene signature for glioma patients. The dataset performed through this study consists of transcriptomic and clinical data found in the Chinese Glioma Genome Atlas (CGGA) (Available online: http://www.cgga.org.cn/, accessed on 8 November 2021) databases [66], as well as of immune cells derived from the TISIDB database. Based on the results of this study, the results showed that glioma patients in the high-infiltration group have worse overall survival than patients in the low-infiltration group. They also claimed that their 15-immune-relevant-gene signature model showed effectiveness and breadth in predicting prognosis in glioma. However, existing limitations of the current article are related to the fact that the impact of other variables related to prognosis (such as age, genetic alterations, WHO grade, and treatments) is not considered, which may have led to misprediction results. Furthermore, the authors made clear that the function of the genes in the signature was a lack of further basic experiments for validation. The immune-related gene signature may help in identifying pathways associated with glioma and potential immune targets for treatment in the future.

Many researchers have taken an extraordinary effort to develop the prognosis prediction model of gliomas, and proposed a survival prediction model of this highly malignant tumor based on MRI radiation characteristics, imaging features from fluorodeoxyglucose-positron emission tomography (FDG-PET), combined with genetic and clinical risk factors. Because of its non-invasiveness and easy access, this field has received more and more attention. In [4], the authors investigated the relationship between tumor shape, quantified using algorithmic analysis of magnetic resonance images, and survival. This study uncovered the relationship between tumor 3D shape and prognosis, which cannot be found only by human flesh eyes. In this study, the researchers gathered each patient’s Fluid Attenuated Inversion Recovery (FLAIR) abnormality and manually delineated enhancing tumor. They implemented a set of features that capture the intricacies of the two and three dimensional shapes and that are independent of the imaging equipment and acquisition parameters. The results showed that a 3D complexity measure bounding ellipsoid volume ratio (BEVR) was strongly prognostic of survival. Lower values of BEVR are associated with poorer survival and indicate a higher level of irregularity of the tumor, which might be associated with a more rapid tumor growth. In addition, three enhancing-tumor based shape features were clinical independent factors. The proposed analysis can be used to help physicians and caregivers customize treatment based on better survival estimates for patients with GBMs. There was also a limitation in this research which is the limited sample size of 68 patients. For machine learning tasks with a small number of samples, a better solution is to apply cross-validation for model selection and preventing overfitting [67].

## 4. Future Challenges

Several challenges must be addressed before the adaptation of machine learning in oncology and specifically, gliomas. Based on the research in the scope of this review, we summarized several main issues listed in Table 3, including data, algorithm, and application. Details are presented in the following sub-sections.

### 4.1. The Challenges in Data

One of the most common issues seen among the studies surveyed in this review was the lack of attention paid to the objective dataset. It is considerable to recognize that several substantive challenges for machine learning glioma analysis at the data level include the (i) lack of annotated data [68]; (ii) ensure data quality and integrity; as well as (iii) data imbalance. 

In the proper training and convergence of machine learning techniques process, the tremendous volume of high-quality and well-annotated data is necessary; however, the multi-institutional nature of most clinical trials for gliomas limit the sharing of patient data between institutions, so the multitude of heterogeneous datasets are difficult to aggregate. Even though large cohorts across many institutions can be aggregated, annotation is a time-consuming process and requires a high degree of expertise. Considering that manual annotations are often costly, the future development of customized semi-automated annotation tools and iterative re-annotation strategies may provide a promising solution by relying on machine learning algorithms to provide initial ground-truth estimates, which are then refined by human experts [69]. In genomics, batch effects [70] often occurs when merging data from different institutions and different batches of experiments. Mean-centering, standardization, ratio-based and EJLR (Extended Johnson-Li-Rabinovic) method [71] are widely used to solve this problem.

As for the second problem, for large datasets data entry and data verification are of essential importance. Further verification or spot checking of data integrity is also a valuable exercise, and implement by a knowledgeable expert. However, in most machine learning papers, the methods employed to ensure the quality and integrity of data are rarely discussed. In terms of data imbalance [72], it is a common issue but has little attention when building and analyzing cancer prediction models. This problem makes the classifier focus on learning most of the data classes so it can poorly classify the samples which belong to the minor class. The paper mentioned above [53] involved in this problem gave a solution. It is feasible for imbalanced data on the high dimensional level to mitigate negative effects through SMOTE [73], adaptive boosting [74].

### 4.2. The Challenges in Algorithm

There are also many tough challenges in the algorithm aspect. For a limited sample size, almost any models are prone to overfitting, which results in an artificially inflated algorithm accuracy [75,76] Although most of the research on cancer prediction models centered on improving the predictability and learning of good representations, they ignored the problem of overfitting due to limited samples as referred by [76]. For example, small samples and large dimensions were particularly big problems for microarray studies, which often have tens of thousands of genes (i.e., features), but only hundreds of samples [74]. In addition, feature selection is a helpful method to prevent overfitting, as shown in previous study [49]. The sample-per-feature ratio is too small to be highly susceptible to overtraining. The problem with overfitting models is that with more and more test cases available, the robustness of models cannot be guaranteed. In overfitting, in addition to striving for large, heterogeneous datasets, L1 regularization is also effective [77], and batch normalization [78] is giving good results in other research. Generally, 5-fold or 10-fold cross-validation is enough to verify most of any learning algorithm [79,80]. Cross-validation is internal validation which is critical to create a robust model that can consistently deal with novel data. In addition to the standard practice of internal validation, it is particularly beneficial to use external data sources for validation testing [81]. The external validation set must be large enough to ensure reproducibility, as well as to help minimize generated objective bias.

The performances of different multiple predictor models is another challenge in algorithms. While sophisticated and advanced neural network algorithms always work well in large heterogeneous datasets, in biomedical tasks complex models are not always the best tools for the task. Overfitting may happen easily on a complex model. Sometimes conventional algorithms, such as support vector machine k-nearest neighbors, and Naïve Bayes can be more popular than sophisticated algorithms, hence their explanatory ability. The assessment of different predictors is a significant step in determining the proper algorithm for the study. Any newly-mentioned model should be compared with the existing models. In our previously discussed research, [53] used random forests and Complement Naive Bayes for comparing, [82] and used random survival forest, SVM, and Cox proportional hazards models to predict survival. However, testing more than one machine learning algorithm is not carried out in most of the papers which we reviewed.

Generalizability of machine learning algorithms is another important issue that should be highlighted here. Most complex machine learning algorithms do not perform well in a small cohort, especially for the large, standardized datasets from multiple institutions with clinical, neuroimaging, and neuropathologic data of gliomas, but up to now most gliomas research focused on relatively small patient populations. Large and heterogeneous populations of gliomas that cover diverse patient populations are needed to fully realize the power of machine learning for the diagnosis and treatment of gliomas. The performance of the model in security and privacy also affects the generalizability, such as Clever Hans. At present, it is much easier to find adversarial examples than to design a model that can defend the adversarial examples. However, there are still some methods to defend adversarial perturbation, such as FGSM (fast gradient sign method) [83] and defensive distillation [84].

The reproducibility of machine learning algorithms is one of the main challenges to transform theoretical research into clinical practice [85]. Machine learning methods, as a computational experiment process like any experiments, are essentially depending on a hypothesis. It follows the defined procedures and needs to verify. Providing the datasets used for training and testing and detailed methodological documentation is of paramount importance. Researchers using open databases such as the Cancer Genome Atlas (TCGA) should at least explain the query used to download the experimental dataset in the Appendix A to permit others to verify and reproduce the results. Parameters of model training should also be stated in detail including how the sets were partitioned. In general, the results of a good machine learning experiment should be as reproducible as other standard laboratory protocols.

### 4.3. The Challenges in Clinical Application

Additional challenges relate to the deployment of machine learning applications in a clinical setting. In early studies of the application of machine learning in gliomas, there is no prospective research to confirm that machine learning can bring benefits to patients’ prognosis. Currently, physicians receive very little training in computer/data science and most computer scientists are not familiar with the complexity of clinical patient management [73]. In addition, image analysis faced many challenges, including the lack of standardization of image acquisition, poor reproducibility, complex quantitative features, and so on [86]. Therefore, comparing results among different institutions may be a challenge, hence limiting the clinical applications.

Another issue is how to understand and trust the information generated by machine learning “black box” [87]. Different from other disciplines, clinical practices can be with a very low fault tolerance rate. The quality of patient life is dependent on the decision maker, which means decisions should be made with a high degree of confidence. Giving meaning to the black box can also form the basis for future work in medical imaging, radiomics, and machine learning.

## 5. Conclusions

This article reviewed the most recent machine learning-based gliomas application models. The considered papers were published during the last 2 decades and used gene expression datasets, as well as medical imaging cohorts, for cancer susceptibility, grade, and survivability risk. This review presented some commonly-used architectures, datasets, and the accuracies of each suggested model. Analyzing the considered papers indicated that machine learning methods can serve as filters, predictors, and classification methods.

It is also obvious that machine learning methods generally improve the performance or predictive accuracy of most scenarios, especially when compared to conventional medical or expert-based systems. Overall, we believe that if the quality of studies continues to improve, it is likely that the use of machine learning classifiers will become much more commonplace in many clinical and hospital settings.

This study has summarized the most recent approaches and their related machine learning architectures. We also highlighted some critical points that have to be considered when building a machine learning-based prediction model, such as reproducibility and data quality. More powerful machine learning-based approaches can be suggested in the future by choosing different model parameters or combining two or more of the presented approaches.

## Figures and Tables

**Figure 1 cells-10-03169-f001:**
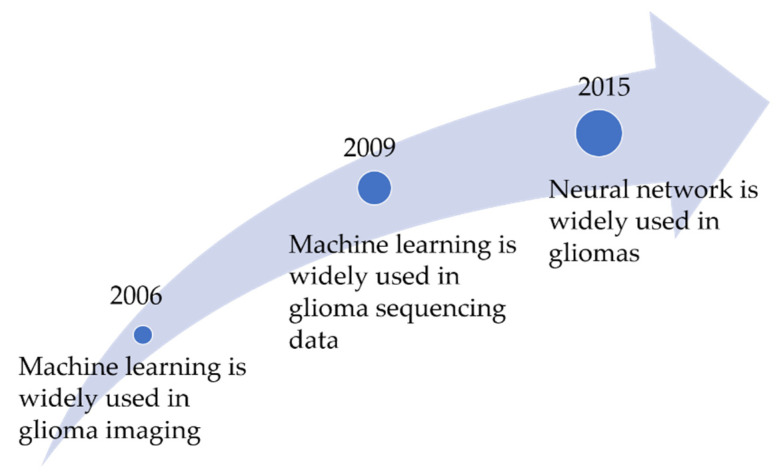
A timeline of the development of the combination of gliomas and machine learning.

**Figure 2 cells-10-03169-f002:**
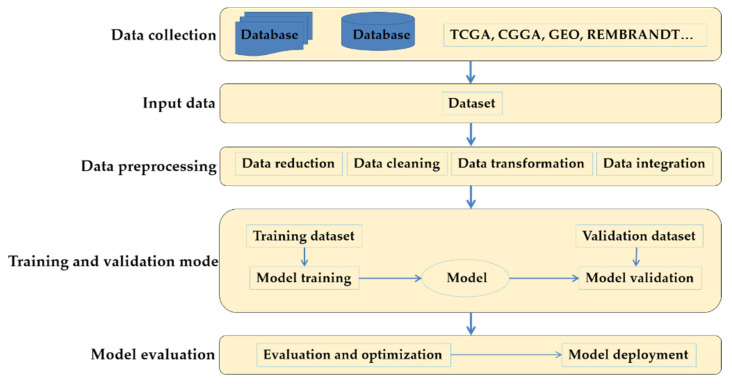
A typical machine learning process.

**Figure 3 cells-10-03169-f003:**
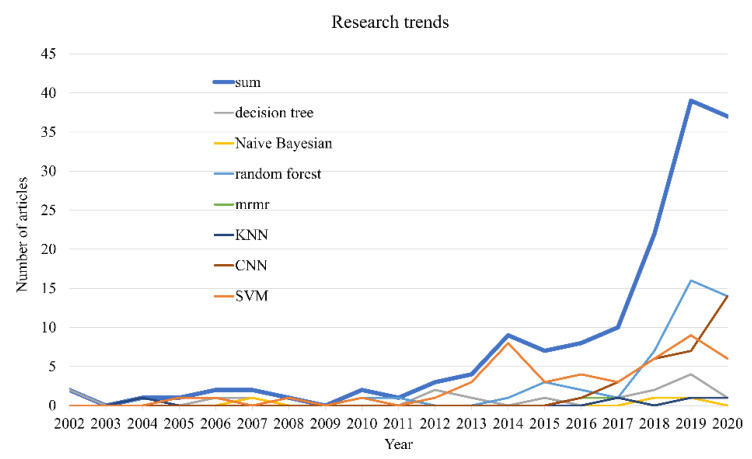
A line chart showing the exponential growth in published papers using machine learning algorithms to solve glioma tasks. The data were collected using a variety of key word searches through PubMed. In this figure, two axes have been plotted. The *y*-axis represents the number for publications related to “glioma” and “machine learning methods”. The *x*-axis represents the publication year. Each line represents the cumulative total of papers published over a year period. The earliest papers appeared in the early 2000s.

**Table 1 cells-10-03169-t001:** Some related articles of other research directions were not discussed in detail in our review.

Category	Reference	Machine Learning Methods	Data Type
feature selection	Zöllner et al. [41]	SVM	dynamic susceptibility contrast MRI
	Sun et al. [42]	L1-SVM + multi-layer perceptron	radiomics
	Abusamra [43]	SVM, KNN, RF and eight other feature selection methods	gene expression data
automatic segmentation	Wu et al. [44]	SVM	T2 weighted MRI
	Chen et al. [45]	multiscale 3D convolutional neural network	MRI
recurrence	X. Gao et al. [46]	SVM	Pre- and post-contrast T1WI and T2 FLAIR
	Rathore [47]	PCA	MRI

**Table 2 cells-10-03169-t002:** Publications discussed in this review.

Category	Reference	Machine Learning Algorithm	Training Data	Year	Aims
Biomarkers prediction	Hsu, J.BK. et al. [48]	random forests	gene expression profile	2019	Identify gene biomarkers
	J. Haubold et al. [49]	linear SVM, random forest	multiparametric 18F-FET PET-MRI and MR Fingerprinting	2020	Identify ATRX, IDH1, and 1p19q status
	Y. Matsui et al. [50]	Neural network	magnetic resonance imaging (MRI), positron emission tomography (PET), and computed tomography(CT)	2020	Identify IDH1, and 1p19q status with multimodal data
Grades classification	C. Lu et al. [51]	SVM and ensemble learning approaches	multimodal MR radiomics	2018	grades classification
	A. Sengupta et al. [52]	random forests and SVM	Conventional MRI images and 3D T1 perfusion MRI data	2019	feature selection before calssification
	B. Niu et al. [53]	random forests and Complement Naive Bayes	gene expression data	2020	imbalanced data problem
Prognosis prediction	P. Mobadersany et al. [54]	convolutional neural network	pathology images and genomics	2018	predict glioma outcomes
	X. Gong et al. [55]	LASSO	Transcriptomic data	2021	develop a signature associated with the tumor immune
	N. Czarnek et al. [4]	Khachiyan and Cox proportional hazards	Axial preoperative fluid-attenuated inversion recovery (FLAIR) and post contrast T1 (T1 + C) images	2017	investigated the relationship between tumor shape and prognosis

**Table 3 cells-10-03169-t003:** Challenges in future research.

Category	Challenges
Data aspect	lack of annotated data
	data quality and integrity
	data class imbalance
Model aspect	Overfitting Clever hans
	lack of comparing with different models
	generalizability of models
	reproducibility of model
Clinical application aspect	Physicians’ knowledge limitations

## Data Availability

Not applicable.

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
