# Peer review of "Research Progress of Gliomas in Machine Learning"

_cells, 2021, doi:10.3390/cells10113169_

Round 1

Reviewer 1 Report

The review summaries the state of the art in applications of machine learning algorithms to gliomas. It covers many aspects of machine learning with respect to gliomas, but misses one point: the real examples when these techniques are used in clinical settings. How accepted in clinical practice these techniques are? Which ones are the most frequently used? Do they really give substantial advantages compared to the expertise of a trained physician? What technical difficulties arise when new techniques are introduced into clinical practice? From the really short section '4.3' it looks like that these algorithms are actually did not get wide clinical acceptance.

Table 1 does not look pretty and besides each query has the same structure with the difference only in the target word. I would recommend deleting it and explaining in the text how publications were retrieved. Another issue is that using only abbreviated keywords you probably missed some publications. For example, you used "SVM" keyword, but not the full form - "Support vector machine". The same goes for mrmr, KNN, and CNN.

What criteria were used to select the publications listed in Table 3? Explain in the manuscript.

It is hard to keep track of the performance of each method. A summarizing table that would contrast the considered approaches would really be helpful for the readers.

As this is review it should be clearly written, so that the readers would not waste they time guessing what the authors meant in a particular sentence. I would encourage the authors to critically go through their manuscript a couple of times paying attention to style and clarity.  See some comments below.

Lines 50-51. "It is the study of computer algorithms that automatically through experience and by the use of data [12]." The sentence is incomplete, correct it.

Lines 90-91: "It mainly uses supervised learning and unsupervised learning to deal with tasks in biology and medicine." From this sentence it is not clear what "uses supervised learning and unsupervised learning ..."

Line 139: search heat has increased fastest. --> search heat has increased the fastest.

Line 192: "Pathological findings are the premise of rational treatment.". "Pathological findings" does not convey the intended meaning. Try to use another expression. Correct it in other places in the manuscript as well.

Line 201: "Therefore, presurgical glioma status and the expression of biomarkers are valued and preferred with non-invasive approaches."

Lines 208-209. Why this particular study? Was it the first one? Perhaps it would be more relevant to describe here the first study that used biomarkers.

Lines 212-215: This sentence is superfluous. It is basically a repetition of the sentence from lines 208-209.

Lines 215-216: "They also mention that when developing prediction models, the combination of multiple biomarkers and the information obtained is of major interest." What information?

Lines 217-218: Combine this sentence with the previous one as this is clearly the "information" that the the sentence on lines 215-216 was talking about.

Lines 218-219: A better formatted sentence would be: In their work, three frequently used machine learning models - Linear regression (LR), SVM, and Random Forests (RF) - were employed ... And do not use hyphens.

Line 224: machine-learning-based radiomics. Do not use hyphens: machine learning based radiomics

Line 234: "...and random forest..." Be consistent: either use upper case of lower case throughout the text to name this algorithm.

Line 241: "robust and diverse committee of linear SVM and random forest". Why committee? Maybe combination?

Line 248: "Another interesting article also published in 2020 with [47]." Drop "with [47]".

Lines 255-257: The sentence needs to be completely restated.

Lines 261-262: "...although it is finally decided by histopathological diagnosis."

Lines 283-284: Restate the sentence. It is poorly written.

Lines 284-286: poorly written. Restate.

Line 311: replace "the affection" to "the effect"

Line 313: noisy expressions --> noisy data

Lines 319-321: This sentence is a repetition of the sentence on lines 315-316. Delete it.

Lines 326-328: restate the sentence. It is poorly written.

Lines 359-360: Provide reference for this statement.

Line 372: "It can tolerant of noisy inputs..." --> "It can tolerate noisy inputs..."

Line 374: replace "support" to "advocate"

Line 376: "Their study redefined criteria..." --> "Their study has redefined the criteria..."

Lines 368-387: In the work discussed in this paragraph histology images and genomics data are integrated. Write a sentence explaining what kind of genomic data was used and how that data was obtained.

Line 400: What "OS" stands for in this sentence?

Line 401: showing --> shows

Line 445: "It is considerable to recognize that ... " --> "It is important to recognize that…"

Lines 449-450: Restate, poorly written.

Lines 462-463: Restate, poorly written.

Lines 486-489: "While sophisticated and advanced neural network algorithms always work well in large heterogeneous data sets, in biomedical tasks complex models are not always the best tools for the job". Explain why.

Line 488-489: "as support vector machine k-nearest neighbors ...". Separate by comma the two algorithms.

Author Response

Dear reviewer:

  We are very grateful to you for their helpful review and valuable comments on our manuscript (cells-1388631). We have carefully revised our manuscript to reflect the points raised by the editor and reviewers. All changes we have made to the manuscript are marked in blue fonts. Please see the attachment.

Reviewer 2 Report

The review with the title "Research Progress of Gliomas in Machine Learning" is handling about the usage of genetic and image information and the usage and publication in the big-data era. The article gives a summary about the current status and the future orientation using Machine Learning approaches. Further, the authors gathered tools and comapre them and are critically wrting about problems like overfitting or class imbalance leading to false predictions.

Introduction:

  1. THe introduction is nicely summarizing from the definition of glioma and the grades over the term MAchine learning and the amount of data. Also the authors already give the intervall of their summary with 20 years. Perhaps a timeline as figure would be beneficial to describe the key tools or marking points for what ML was now used in Glioma research.
  2. Solely one point could be more described. Missing is in the introduction now the conenction of ML and Glioma research. So which medical questions are answered or should be answered by ML using which input data. SO giving something like an explanation of prognosis detection using image data or genome variations to diagnose early the risk or something like this.

THe Machine learning Algorithms:

  1. THe first paragraph is redundant to the introduction and should be removed here.
  2. THe figure is of very low resolution and the font size to small.
  3. THe phrase of main machine learning techniques and then relating to decision tree and Naive BAyes is not quite true in general. It dependes on the question and field of research what is a main ML model. SO please remove/rephrase this statement.
  4. The Authors should also consider simple neural networks or ANN (Artifical neural networks) and not only the special case of CNN. Also Table1 is good but more for a supplementary as it is very long and not necessary to udnerstand the flow of the review.
  5. Missing is in this section more what data are used for what ML approach or what were the questions answered with the different methods. THe cross-validation paragraph comes out of nowhere and should be removed here.

A Survey of MAchine learning Applications in Gliomas

  1. This chapter is stil missing the connection to the glioma data and predictions. There are some medical applications mentioned but table 2 is not containing the mentioned applications like progression, tumor drug sensitivity or other diagnsis parameters. Mentioning there three tools and all based on SVM without clear relevance is not good in a review giving an overview of the current status in the field. THere is not only one publication for feature selection available.
  2. Table 3 is in this sense better but missing is as stated in the introduction the 2 decade interval. THis should be visualized here in the review.
  3. The paragraph should more focus on the used biomarkers and inputs. At stage the sections is quite unclear without a clear line. For example the MRI data are mentuoned but not clearly what are now the marker to define a glioma or the grade of glioma. Also sequencing data are not mentioned at staage here as a potnetial biomarker.
  4. Also before section 4 already some of the paragraphs are talking about limitations and problems. Here I would like to see a more laymen strcuture. So the Challanges and Drawbacks in section 4 and the clear approaches in Section 2. IN section 3 for example now the different biological questions answered in the respect.
  5. THe challenge section contains very few references at stage and could be more extended to problem fiels like Balancing, Clever hans predictors, Overfitting, BAtch effects and noise.

Overall Missing is the more direct focus on Glioma and what was achieved to glioma research and diagnsosi with the help of ML. It reads at stage more like ML and from time to time a cross reference to glioma. THe authros should think about clear structuring and adding more schemes or timelines or ML approach schemes for clarity. Overall the review is valuable but also references to Pan-Cancer studies could be beneficial. In the end the focus of the paper is more on the last 3 years and not 20 years as written. THis should be more from the historical point of view be considered.

Author Response

(The authors gave the same response as above.)

Reviewer 3 Report

The whole article is interesting and very well thought out and written. It is a bit long to read and could be shortened. This comment is mostly about the first part, it would be good to put a good part of it in addition. For example, Table 1 is not essential. Also, the questions that deal specifically with approaches and measures could be reduced.
The space gained could be used to bring in a little more criticism and evaluation of the work; this is missing in this excellent article.

Author Response

(The authors gave the same response as above.)
